# Hearing the voices of midwives through reflective writing journals: Qualitative research on an educational intervention for Respectful Maternity Care in Zimbabwe

Sunanda Ray[1]*, Christina Mudokwenyu-Rawdon[2], Myriam Bonduelle[3], Ginny Iliff[4], Caroline Maposhere[2], Priscilla Mataure[5], Cecilia Jacobs[6], Susan C. Van Schalkwyk[6]

1 Department of Community Medicine, University of Zimbabwe College of Health Sciences, Harare, Zimbabwe, 2 White Ribbon Alliance for Safe Motherhood, Harare, Zimbabwe, 3 Swansea Bay University Health Board, Swansea, Wales, United Kingdom, 4 Private obstetric practice, Harare, Zimbabwe, 5 Department of Family and Health Sciences, Women's University in Africa, Harare, Zimbabwe, 6 Faculty of Medicine and Health Sciences, Centre for Health Professions Education, Stellenbosch University, Stellenbosch, South Africa

* sunanda28@hotmail.com

**Data Availability Statement:** There are ethical restrictions which prevents the public sharing of

## Abstract

Women attending public and private sector health facilities in Africa have reported abuse and neglect during childbirth, which carries a risk of poor health outcomes. We explored from the midwives' perspective the influence of an educational intervention in changing the attitudes, behaviour and practices of a group of midwives in Zimbabwe, using transformative learning theory as the conceptual framework. The twelve-week educational intervention motivating for Respectful Maternity Care consisted of a two-day workshop and five follow-up sessions every two weeks. Thematic analysis was conducted on eighteen reflective journals written by the midwives with member-checking during follow-up discussions and a further one-day participative workshop a year later. The midwives reported being more women-centred, with involvement of birth companions and use of different labour positions, stronger professional pride and agency, collaborative decision-making and less hierarchical relationships which persisted over the year. Their journal narratives included examples of treating birthing women with more compassion. Some categories aligned with the phases of transformative learning theory (self-examination of prior experience, building of competence and self-confidence into new roles and relationships). Others related to improving communications and effective teamwork, providing role-models of good behaviour, use of scientific knowledge to inform practice and demonstrating competence in management of complex cases. This study shows that innovative educational initiatives have the potential to change the way midwives work together, even in challenging physical environments, leading to a shared vision for the quality of service they want to provide, to improve health outcomes and to develop life-long learning skills.

minimal data for this study. Data will be retained for storage and safekeeping at the Centre for Health Professions Education, Faculty of Medicine and Health Sciences, Stellenbosch University (www. sun.ac.za/chpe). The data will be specifically stored within a password protected folder which can be accessed by current and future CHPE Directors. Data are available upon request from Lorraine Louw, Postgraduate Program Administrative Coordinator of the CHPE, Director of the Centre for Health Professions Education (acting), Stellenbosch University via mail (Centre for Health Professions Education / Faculty of Medicine and Health Sciences / Universiteit Stellenbosch / PO Box 241, Cape Town 8000 / Francie van Zijl Drive Tygerberg, 7505 / South Africa), email (lhl@sun.ac. za), or telephone (+27 21 938 9047), for researchers who meet the criteria for access to confidential data.

**Funding:** Author MB was the recipient of a grant of £20 000 from the Welsh Governments' Wales for Africa Grant Scheme for funding the RMC workshops and production of the RMC toolkits Project reference no: WFA Rd3_17. Website: https://www.gov.wales/wales-and-africa#section-14799 No other funding was received for this project by any of the authors. The funders had no role in study design, data collection and analysis, decision to publish, or preparation of the manuscript.

**Competing interests:** The authors have declared that no competing interests exist.

# Introduction

Women in public and private sector maternity services in Africa have reported a wide range of physical and emotional abuse and neglect from health professionals (including midwives) in well-resourced and poorly-resourced settings [1–12] including in Zimbabwe [13]. The consequences of scolding and controlling approaches towards women in labour include poor obstetric outcomes such as delayed progress in labour, higher risk of instrumental and surgical interventions and postnatal depression, and are often unrecognised [14,15]. The dissonance between midwives' professional pride and their apparently disrespectful behaviour in their workplaces creates tensions between them and the women in their care. Negative birth experiences may deter women from seeking institutional care during labour, whereas increasing rates of skilled birth attendance and facility-based childbirth are considered vital to reducing maternal morbidity and mortality [2,3,9,11,12].

A review of relevant literature, documents and interviews identified seven categories of disrespect and abuse in childbirth: physical abuse; non-consented care; non-confidential care; non-dignified care; discrimination based on specific patient attributes; and abandonment of care and detention in facilities [11]. These categories of abuse have been found from research in health facilities in African countries to be more widespread than anticipated, dispelling the notion that acts of abuse are isolated events carried out by "bad apple" individuals [1,2,4,5,8,11]. The failure of authorities to trigger disciplinary action in response to abusive activities has "normalised" such behavior especially in the eyes of health staff and is an indicator of health system weaknesses and failure of accountability [3,4,8,9]. The missing voices of midwives in research on abuse of women in childbirth has resulted in lower appreciation of the work constraints within which most midwives' behaviour is embedded, which some argue constitutes disrespect for midwives themselves [15].

Research from African countries has shown that complex relationships between midwives and birthing women give rise to dissonance in the professional identity of midwives [8,9]. Midwives are expected as professionals to be altruistic, caring and to put the interests of women in their care first, but instead may demonstrate disrespectful behavior. They may purposefully use abuse to create social distance between themselves and birthing women, based on their perceptions of class, power, identity and knowledge superiority, as a protective strategy to avoid engagement and to preserve the power differential between them [8,9,16]. Professional ethics that champion women's rights to information and autonomy could appear threatening to midwives in this context [8]. Medicalization of childbirth, with its emphasis on technology and pharmacological interventions, gives midwives control with rules and discipline over the experience of childbirth, rather than allowing women to have autonomy, to take positions they are comfortable in or by having companions to support them [6,15]. Some midwives who may be fearful of being blamed for poor outcomes, attempted to control women during childbirth, including by punishing them for disobedience with physical abuse [15,17]. This disrespect, neglect and abuse has been termed obstetric violence by some authors [14,17,18]. During institutional births, especially in low-resource settings, women may be left to give birth alone and are vulnerable without companions who could mediate for them and provide support [6,7]. Continuous practical and emotional support from birth companions during childbirth is associated with positive experiences and improved outcomes for pregnant women and their partners [14,19].

Abusive behaviour towards birthing women has mainly been studied using exit interviews and focus group discussions with service users and less commonly, with providers [1,2,10,11,15]. Various interventions have been proposed to motivate for more respectful behaviour. These include stronger nursing leadership, professional accountability, training to

promote respect for patient autonomy, better adherence to codes of ethical practice, quality improvement initiatives, inclusion of birth companions and creating confidential complaints procedures that will not lead to patient victimization [1,7,8,11]. The legacies that have shaped health systems, the models of care and training, cultural norms within which they have developed, and the perspectives of midwives require greater scrutiny. Enabling midwives and pregnant women to develop greater agency and advocacy skills, despite being apparent victims of restrictive institutional cultures and health system failures, could lead to improvements of those systems [6,9,15].

The Respectful Maternity Care (RMC) Charter, which was developed in 2011 by a group of international stakeholders, promotes the rights of women receiving maternity services [20]. Key principles of RMC are freedom from harm and mistreatment, privacy and confidentiality, the right to dignity, to information and effective communication [20,21]. Compassion, the increased attention to the suffering of others with intention to assist them [22], is an important component of RMC. Interventions achieving RMC are expected to empower midwives and women in childbirth, giving both more agency and control in making decisions for themselves. For midwives this includes improved morale, greater professional pride and relatedness to each other. Building resilience, the ability of individuals to respond positively to adversity using effective coping strategies, is another component, based on midwives sharing social support and developing self-caring skills [23,24]. Education of midwives on the emotional and physiological aspects of labour, would expand their knowledge base to include understanding how scolding and shouting at women contributes to poor maternal and newborn outcomes [14,15,23]. However, education alone is insufficient to change attitudes and behavior and needs to be accompanied by a process of life-long learning that feeds into the construction of their professional identity as midwives.

Frenk et al (2010) have motivated principles of transformational learning (rather than informative or formative learning) as essential for strengthening health systems and improving quality of care [25]. The World Health Organization has also advocated more positive birth experiences [26]. Our interest was in how to put these concepts into practice. This qualitative study engages with the phenomenon of advancing RMC from the midwives' perspective, to encourage sustainable behavior change, using methods grounded in promoting professionalism and rights of birthing women. The objective of the study was to assess changes in attitudes, behaviours and practices in a group of midwives in response to an educational intervention promoting RMC. Mezirow's transformative learning theory [27–29] was used as a conceptual framework to explore the learning journey these midwives made in becoming "enlightened change agents" in relation to their work and for the birthing women they served. Critical self-reflection and challenge of core beliefs, assumptions and values, as a means of acquiring new knowledge by adult learners, are central elements of transformative learning theory. The ten phases of transformative learning [29] are listed in Table 1. Events of dilemma or dissonance are turned into opportunities for reflective inquiry, using for example dialogical learning within a group of midwives, to explore how they feel about incidents with women in their care [30,31]. Learners revisit their values and beliefs in relation to their new experience, questioning whether their worldview is accurate, to be changed or preserved. They discover new insights and self-awareness in relation to others. If learners change their views without questioning, or adopt new beliefs because of pressure on them, it will not be transformative [27].

We chose to use reflective writing journals as a source of data on the experiences and learning of the midwives in this study. Diaries and journals often provide better data for research than interviews or observations, especially where the information being gathered is sensitive or revealing. They are less limited by recall bias since entries are contemporaneous [32,33]. Reflective narrative writing by health professionals is a skill that increases self-confidence in

**Table 1. Alignment of changes in midwives' attitudes, behaviours and practices with the ten phases of Mezirow's Transformative Learning Theory [29].**

| Ten phases of transformative learning | Analysis of midwives' journals and dialogue |
|---|---|
| 1. disorienting dilemma | The reflections started with disorienting dilemmas as per the Gibbs reflective writing cycle [40] which starts with "what happened?" |
| 2. self-examination with feelings of guilt or shame<br>3. critical assessment of epistemic, socio-cultural or psychic assumptions | Domain 1: **Changes in attitude**<br>• self-examination of prior experience: their feelings of anger and frustration in the past changed on reflection to regret, guilt and shame<br>• reconsideration of assumptions<br>• positive feedback as motivator<br>• perception of capabilities and application of specialist skills as midwives<br>Recognition that they had been part of a health system that abused women through conflict and neglect, especially verbal abuse and by ignoring their needs, where the rights of birthing women were not respected, and that the disrespect went all the way to cleaners and security guards |
| 4. recognition that one's discontent and the process of transformation are shared and that others have negotiated a similar change.<br>5. exploration of options for new roles, relationships and actions<br>6. Planning a course of action<br>7. Acquisition of knowledge and skills for implementing one's plan<br>8. provisional trying of new roles | Domain 2: **Changes in Behaviour**<br>• Manifestation of resilience and compassion<br>• Communications, teamwork and role models<br>• Competent management of complex cases<br>Recognition that there were small incremental actions they could take to improve quality of service despite the deficiencies of the health system.<br>Revisiting their core beliefs and values took the midwives back to their shared vision of what being caring professionals meant.<br>Collective exploration through dialogue of options for new ways of interacting with birthing women and with each other<br>Peer pressure to behave differently<br>New knowledge from training on theoretical aspects of childbirth, interrelationship of hormonal, psychological and emotional influences on labour and positive outcomes, endorsed changes in behaviour<br>Increasing agency in the decisions made |
| 9. building of competence and self- confidence in new roles and relationships | Domain 3: **Change in practices**<br>• Involvement of birth companions in labour and giving birth<br>• Encouragement to use different positions during labour and birthing<br>Progression into new roles and relationships that were more empowering for themselves, birthing women and colleagues.<br>Good outcomes from new practices reinforced the self-confidence and self-efficacy they required to sustain their skills.<br>Collective acknowledgment of their expertise, through shared experiences of what they did well, recognising that mistakes they made were also opportunities for learning for each other and individually |

(*Continued*)

**Table 1.** (Continued)

| Ten phases of transformative learning | Analysis of midwives' journals and dialogue |
| --- | --- |
| 10. reintegration into one's life based on conditions dictated by one's perspective | Key changes in practices sustained over time, despite workspace limitations, such as encouraging women to move around during labour and to adopt positions of comfort for themselves while giving birth; attendance of birth companions in private and public sectors where possible while allowing for privacy for other women; giving information to women in labour and their families, involving women in decision-making, behaving as responsible, trusted professionals to create an environment in which women felt safe and respected. Positive professional identification as RMC champions and patient advocates, determined to disseminate new learning to colleagues and to improve the public perception of midwifery as a compassionate profession and discipline.<br>Midwives spoke of creating time each day to reflect on the cases they had worked with and how to improve their management. More discussions about difficult cases and lessons learnt occurred during team meetings which took place more regularly. |

applying knowledge and skills, allowing them to critically assess the impact of their behaviour, which when connected with bearing witness to events of tragedy and wonder, may modify their behaviour towards more patient-centred care [23,31,34,35]. Collaborative reflective practice through journal writing and dialogue between midwives could lead to more nurturing relatedness with each other and women in their care. Their sense of self-efficacy and professional identity would derive from belonging to a community of practice, which awakens their sense of resilience. Communities of practice are networks of individuals who, through their interactions together, develop a common sense of identity, sharing knowledge, exchanging ideas, insights, values, stories, practices and expertise [36,37]. Revisiting core beliefs and values through this process would take midwives back to their understanding of what being a caring professional means. Critical reflection skills, self-awareness and effective life-long experiential learning are necessary for remaining competent as health professionals who are able to self-monitor and self-regulate [35,38].

## Methodology

### Ethics statement

Ethical approvals were granted by the Stellenbosch University Human Research Ethics Committee (S19/02/042) and the Medical Research Council Zimbabwe (MRCZ/B/1743). The midwives who submitted their journals for analysis and those who attended the workshops gave written consent for their reflections to be included in the study, on condition that their contributions would be anonymized and neither they nor their workplaces were identifiable.

This qualitative study uses a phenomenology approach, exploring the first-hand accounts of the lived experiences of individuals, and situated within the interpretive paradigm in which the contextual features of their experiences are interpreted in relation to other influences such as culture, social norms and identity [39]. The methodology comprised a document analysis of eighteen reflective journals written by the midwives during an educational intervention conducted in 2018, followed by member-checking of the analysis of the journal entries during a follow-up session and a one-day participative workshop held a year after the initial educational

intervention was completed. The educational intervention motivating for RMC consisted of a two-day workshop with theoretical inputs (including the RMC Charter) and small group discussions, with five follow-up sessions every two weeks, altogether a twelve-week programme. The facilitators for the educational intervention and co-authors of this paper were the researcher (SR), a public health physician; two senior midwives (CM-R and CM) who initiated the RMC training through the White Ribbon Alliance Zimbabwe; an obstetrician working in Harare (GI) and an obstetrician working in Wales (MB) who had previously worked in Harare. A third senior midwife (PM) joined to support the process with a view to incorporating RMC into midwifery curricula at the Women's University Zimbabwe. One midwife trainer from Wales facilitated during the initial workshop and a second facilitated at a later workshop, though are not co-authors. The book *Supporting women for labour and birth, a thoughtful guide (Leap and Hunter, 2016)* [23] was given to each participant as a guide for the training.

Midwives in Zimbabwe are qualified professional nurses with at least one additional year of midwifery training. Their language of instruction is English. The midwives who participated in this study were from public and private sector maternity facilities in Harare (capital of Zimbabwe). They were selected by their employers in response to a written request for experienced and highly motivated midwives. A purposive sample of twenty-four midwives attended the initial workshop, while twenty attended the follow-up sessions, of whom eighteen submitted their journals for review. The midwives hand-wrote in their journals in English about significant incidents that happened during work, using the Gibbs reflective writing cycle [40], explained to them at the initial workshop. They described each incident ("what happened?"), their feelings about it, how they made sense of what happened, what else they could have done, and lessons they learnt for future action. They were also encouraged to narrate what went well in their management of incidents. We collected the journals at the fourth follow-up session and conducted an initial analysis which we presented to them at the fifth session, with anonymized narratives and quotations from their journals. A one-day participative workshop followed a year later, when the midwives discussed the changes they had been able to sustain during that year. The data generated during these feedback sessions were collected from rapporteur notes and flipcharts where key points from the conversations were recorded.

## Data analysis

The journals were reviewed using document analysis, a systematic analytical process of selecting, making sense of and synthesizing data contained in documents [41]. The researcher (SR) read through all eighteen journals and developed a framework for analysis, manually identifying codes and assembling them into categories within the domains of attitudes, behaviours and practices which were determined in advance of data collection. These were discussed and verified with the midwives and facilitators following the presentation of the initial analysis during the fifth session. The journals were reviewed again, with modification of the categories through interaction with the data from the workshops and the one-year follow-up. The findings from the analysis of this data are given below.

## Findings

Analysis of the midwives' experiences showed how changes in their attitudes, behaviour and practices aligned with transformative learning theory (summarized in Table 1) and arose through the learning process. The midwives started by describing in their journals, incidents at work which were challenging to them ("disorienting dilemmas") such as women presenting with haemorrhage or sepsis following a backstreet abortion, a baby still-born, a woman screaming in pain, or a quarrel with an aggressive colleague. The midwives continued in their

narratives, making sense of their experiences, confronting previous assumptions and learning through interpreting their discoveries and dialogues with each other as active participants. They gave examples where they had successfully applied new knowledge or understanding and how they felt about themselves as a result. The data from these journals and dialogues was coded into the following categories within the three domains of changes in attitudes, behaviour and practices.

## Changes in attitudes

**Self-examination of prior experiences.** The midwives described how in the past they felt anger, frustration and exasperation when women in labour appeared unresponsive or uncooperative, for not doing what they were told. Some had been taught that women did not progress in labour ("*were lazy*") if midwives were nice to them so were surprised to learn scientific evidence for the opposite. Several disclosed feeling regret, guilt and shame, especially for shouting at women instead of supporting them, recognizing that such responses could traumatise birthing women psychologically.

> *I felt ashamed of what we do as midwives at times. . . the experience and circumstances surrounding the labour and delivery process, demotivates and dehumanises (women in labour)* (mw12).

> *I also have bad obstetric history. . . I knew how she felt when we rushed her baby for resuscitation. . .her tears burned my own cheeks and I went to the toilet to cry. . . I have to be a better advocate for my patients. . .in future I will prioritise such cases and press the obstetrician to act swiftly* (mw15).

When one midwife did not follow her instincts to avoid an unnecessary caesarean section, the depth of regret she recorded made her determined to be more assertive if a similar incident happened again.

> *I felt a failure. . .I regretted I had not been persistent and lacked the confidence to stand up to the other midwife. . . next time I will be vigilant and stand my ground* (mw16).

They acknowledged providing inadequate quality of care in the past and could see the RMC training leading to improvements. Through rehearsing techniques they learnt in the workshop, they cast themselves in a new light, as self-aware, caring, knowledgeable, competent health professionals, treating women with respect as patient advocates, change agents and RMC champions.

**Perception of capabilities.** The narratives revealed joy in their new relationships as competent health professionals and good role-models for their colleagues, increased self-confidence and pride in their work: "*I am one of the best midwives in my unit*", "*pleased and satisfied with what I have done*", "*inner joy*", "*positive energy*", "*humbled to bring a life to the world*", "*I feel an achiever, happy and motivated*". Receiving compliments and appreciative feedback from colleagues and women in their care motivated them to do better. Mw16 described how a patient thanked her saying, "*you are young, but you know how to take care. . . you gave me a sense of security and confidence*" (*mw16*). Others said, "*(My colleague) told me she liked my attitude towards work and patients*" (mw15); "*The appreciation of women post-delivery gives me the encouragement to do more every day*" (mw12).

The midwives praised women for being strong and coping well during labour, encouraging them to be more communicative and responding to questions with more information and

guidance, instead of impatience. A key change was in listening to pregnant women, hearing their stories, and sympathising with their difficulties rather than judging them negatively. These changes in midwives' attitudes, of increased self-confidence and perception of their capabilities, connect with their competent management of complex cases described later. They described being more assertive through feeling knowledgeable, skilled and competent, exercising their professional judgement. When these initiatives were successful, their pride and confidence led to a greater sense of positive professional identity, where they could observe through reflection the impact of their behaviour on improving birth outcomes and quality of care.

**Reconsideration of assumptions.** Better understanding of the motivation (or lack of) of women during labour was key to supporting changes in midwife perceptions and attitudes. The "uncooperative" woman was judged to be disobedient or unconcerned about their health or the wellbeing of their babies. These assumptions were challenged when they learnt through each other's narratives how fearful and anxious women were, especially after hearing distressing childbirth stories from others. Negative memories from previous painful deliveries also contributed to fear. Learning the science of hormonal responses to fear revealed to them how women's emotions could retard and shut down labour. Instead of bullying, they learnt to listen, to consider the home circumstances and financial constraints faced by women in their care, explanations for why they delayed booking for antenatal care or arrived late already in labour, why they travelled to a clinic when in labour instead of to a hospital as instructed. Stories were told of women who came from remote areas without money for booking or transport, who were pregnant when they did not want to be and abandoned when the pregnancy was revealed, with no family to support them. Through dialogue they began to understand where their anger and exasperation came from, that by treating women with disrespect, they were disrespecting themselves as women. They expressed sadness for experiences of women in their care which made them behave in ways that the midwives would have previously found unreasonable and annoying. Several journal entries described young women, usually in their first pregnancy, fearful of being examined during labour or sutured following birth. These were often women traumatized and abandoned in their relationships, non-attenders for antenatal care, who did not know what to expect during delivery. In this narrative a teenage woman in labour refused to be examined, despite explanations about the need to assess her progress:

> When I tried to examine her, she refused . . . I tried everything. . . I decided to stay with her . . . let her to do whatever she wanted. . .when I saw she was going to deliver. . . .I had to call other midwives to help and we finally delivered a live infant. (mw6)

While discussing this story, the midwives reminded each other that sexual violence leading to pregnancy can traumatise women so they can't bear to be examined, so greater sensitivity was needed in supporting them in giving birth. The midwives described previously pushing distressed women away because they felt helpless when faced with their troubles. They were now eager to make women's birthing experiences fulfilling even when they could not change other difficulties in their lives.

> Understand the full story, never judge. . . there is always a story behind the mother, her situation, her social history, her physical and mental health (mw8).

> Previously I would have cut her short, saying . . . you are wasting my time. Because of the RMC training, I gave her time to air out all her problems. . . when we finished, she thanked me and went away happily (mw5).

These reflections displayed better understanding and listening skills, attributes which enabled them to provide improved quality of care.

## Changes in behaviour

**Manifestation of self-care, compassion, resilience and professionalism.** The midwives narrated stories showing more compassionate behaviour towards themselves, each other, and birthing women. Several wrote about the emotional exhaustion they experienced during stressful times, resulting in rudeness, scolding and shouting as they transferred their frustrations to women in labour. They described instead using coping and self-care skills learnt during the RMC workshops, which enabled them to be resilient in adverse circumstances. These skills included deep breathing exercises, keeping calm, stretching, singing quietly, talking in slower lower tones, being purposefully kind and smiling more. They reassured the women, rubbing their backs and explained to them what was happening. Their new experience of responding sympathetically to the crises birthing women presented with and of forming new relationships with them, was rewarding and fulfilling, showing them ways to improve women-centred care. They understood better the origins of their anger and frustration towards women in labour, especially young vulnerable women. A midwife narrated in her journal how a 17-year-old teenager who gave birth to a live pre-term low birthweight infant, wished the baby had been born dead; she was not ready to be a mother and had been abandoned by its father. The baby died in the ambulance on the way to hospital. The midwife commented:

*I would have shouted and scolded the girl, why she didn't book . . . why she had kept the pregnancy a secret to the family. . .I would have blamed her . . . why she slept with an unknown person without protection and got pregnant. . . (following RMC training) I treated her with love and caring, explaining everything that was happening, the reason for the transfer. . . (mw14).*

The midwife wrote that she now felt compassionate towards the young woman, guided her on future family planning and advised her to confide in her sister about what had happened. Another midwife described how a 17-year-old attended the clinic with heavy bleeding and abdominal pain following an induced abortion, initially refused transfer to hospital then changed her mind.

*Because of the respect I gave her despite her age she discussed everything with me which made my care for her easier (mw5).*

The midwives discovered they could alleviate the anxiety of women in their care with explanations, reassurance, and making them feel safe, which facilitated labour. When a woman at risk of antepartum haemorrhage gave birth vaginally because the doctor was held up, the good outcome was a lesson to all involved.

*The woman was very thankful, she said, 'I was afraid but you were with me all the way. . .even when you were talking to the doctor on the phone I could hear your concern so I knew I was surrounded with people who care' (mw16).*

The midwives came to see that giving information, rather than withholding it, was a powerful instrument for bonding with women in their care, helping them understand their condition, and for working together towards better health outcomes.

**Communications, teamwork and role models.** The midwives learnt how to collaborate and communicate better with colleagues, women in labour and their families, developing skills

in advocacy, persuasion and negotiation. They explained to women why they needed to be transferred to the next level, why they needed procedures, why something had gone wrong like a baby's death. Working as teams became essential to their sense of resilience, to "*be there for each other so that women feel safe, loved, and cared for*" (mw14).

> *If a midwife collects such a history (HIV positive teenage pregnancy with poor family circumstances) it is important to tell the next midwife (at handover) so that there is . . . continuity of care for that woman" (mw17).*

They self-identified as change agents in motivating improved practices in colleagues who had not received RMC training, encouraging each other to behave professionally, to control their emotions and be calm.

> *I found (the obstetrician) shouting at a patient. . .I asked him to take a break. . .(and) talked to him about RMC. . . how I would like to be treated if I were a patient. . .He promised to perform better and smile more (mw15).*

The midwives shared their stories and experiences as team-players, learning collectively how to improve their practice and acting as role-models of professional behaviour and RMC champions.

**Competent management of complex cases.** The midwives previously acquiesced to the authority of senior colleagues, even where they disagreed with their opinions, because of their sense of lower social status. They felt their clinical judgement was disrespected, especially by some obstetricians, that they were expected to only carry out instructions: "*for too long midwives have acted as the doctors' handmaidens*" (mw8).

> "*Some obstetricians respect midwives because . . . they have more time with patients. . . and act in their best interest. They work as a team. . . may ask the midwife for their opinion and respect (it). Other obstetricians look down on midwives, they feel they . . . know best, and the midwife must follow suit*". (mw2)

Following the RMC training, they reported taking more initiative, gaining confidence by applying skills in managing complex cases, advocating for women in labour, responsible to protect their health and well-being, and in resolving difficulties using their professional judgement. During the follow-up sessions, they advised each other on conflict resolution and negotiation. Two midwives described feeling proud and competent when their thorough physical examinations alerted them to the risk of uterine rupture and haemorrhage in women in labour, leading to emergency operations. Their judgments were endorsed by senior colleagues, which strengthened their professional pride as midwives. In the following case the midwife negotiated with her manager to intervene with an obstetrician.

> *I phoned the doctor . . . he was reluctant to come. I informed the manager-on-duty (who) phoned him to come to see the patient. They took (her) to theatre . . . the patient had a ruptured uterus. A total hysterectomy was done and the bleeding arrested. . . a life was saved (mw11).*

Another midwife questioned why a woman was still bleeding heavily after birth, though the notes stated complete delivery of the placenta. After initial interventions failed, she resorted to manual evacuation of retained tissue in the uterus, putting up an oxytocin infusion. The

woman stopped bleeding and recovered. In her journal the midwife stressed the importance of making her own assessments and of passing that learning on to her colleagues.

*I was very happy. . . (to) have done something by focusing on (the woman's) condition, rather than the patient being sent to the postnatal ward and complicating there. I discussed . . . with my colleagues and the doctors, not to take postnatal examinations for granted. . . (we) should not rely on what has been documented, since people are bound to make errors at times (mw16).*

Occasionally distressed parents of a stillborn baby or one born with malformations, would direct their anger at the midwives. The midwives reflected that they learnt to remain calm, not to be defensive or angry, to encourage parents to express their grief. They accepted that they were occasionally unprepared for tragic scenarios. One midwife disclosed that midwives needed to address their own fears to counsel women appropriately:

*We. . . often overlook the pain and sadness women feel . . . when babies die, we show the body to the mother just for a short time and it is taken away from them when pain and grief is still very raw. . . we cannot deal with women's fears without starting with ourselves. (mw6).*

They reflected on the vulnerability of women during childbirth, that negative experiences could affect them and their babies for a long time, of the importance of gauging the emotional and psychological status of birthing women and to identify possible sources of assistance for them.

**Changes in practices.**   During the RMC training, the midwives learnt the psychoneuroendocrinology of childbirth, about how continuous emotional and psychological support helped labour to progress and is the basis for encouraging birth companions. The training provided the rationale for women in labour to walk, dance and sway, choosing positions they were comfortable in during contractions and giving birth. Video clips and simulations were used to demonstrate how this could be done. Previously it was rare for birthing women in public sector facilities to have companions and they were usually instructed to lie in lateral positions on their beds during labour and gave birth on their backs.

**Encouragement to use different positions during labour and birth.**   The midwives described encouraging women to walk and use positions they were comfortable in, though they had to learn how to examine them in different positions. In their journals and during the follow-up discussions, they told stories of women being assisted to give birth standing, on all fours on the floor or on the bed, squatting, kneeling.

*A woman in labour was shouting at the top of her voice saying she was in pain. . .she decided to sleep under the bed. . .there was no room to do the monitoring due to space limitation. . . (we) had to kneel on the floor to support the perineum. . . the baby came out on the floor (mw17).*

She wrote that in the past the midwives in labour ward were going to "*exchange words*" with this woman, that this behaviour was unacceptable. She reflected that the RMC training helped them to accept that every woman behaves differently in labour and their role is to accommodate her needs as far as possible. Others said:

*The woman preferred walking around. . .(she) was very cooperative and 'breathed' the baby out rather than pushed. . .(her) perineum was intact. . . I (wished) I could have called all the*

*midwives who were present to witness. . .but everything happened so fast. Now I am confident to use the position and teach others how to deliver in that position (mw15).*

*Each time I did a vaginal examination and explained to the woman what I was doing, it motivated her to do more, walking around, rocking, sitting in the chair and deep breathing exercises during the contractions; she did very well and delivered a baby boy . . .she was just full of joy (mw12).*

The midwives were encouraged by observing how the techniques they learnt during the workshops facilitated the progression of labour.

**Involvement of birth companions in labour and birth.**   The midwives described how they now accepted women relatives attending women in labour, one at a time. In private sector facilities with more privacy, husbands were also welcomed. They witnessed the benefits when companions provided emotional and physical support, releasing midwives to carry out other tasks, especially when looking after several women in labour at the same time.

*She did not want me to leave her, calling me to rub her back each time she got a contraction. . .then called her young sister to come to rub her back as her companion . . . (mw2).*

*We allowed her to be on her (hands and knees) on the floor on a drawsheet with the husband doing backrubs (mw13).*

They appreciated the benefits of companions giving explanations and encouragement, creating a calmer, more relaxed atmosphere.

*The presence of her husband changed the situation completely, she cooperated better and I sutured her with no problems (mw15).*

*The husband. . . was very supportive, encouraging the woman to try what they were taught, kneeling on the bed and walking around. When labour had not progressed well. . . the woman requested . . . the mat so she can lie on the floor. . . with the husband sitting by her side, she delivered (mw12).*

Companions also helped bereaved women to cope, such as when a mother could not look at her dead baby but was supported by her sister who viewed it for her (mw9).

**Structural challenges to RMC.**   This study was not specifically designed to explore structural and policy-related factors that impact the quality of maternity care. Some key responses emerged however, on the ability of this group of midwives to practice RMC within limited resources. The midwives described problems related to their busy workloads, not having enough space to provide individual care, with public sector services being overwhelmed and crowded, how these affected the quality of care they provided:

*I had to deliver a patient in a cubicle with two other patients. . . I felt so helpless, unprofessional and disrespectful to (my) patient. . .I (asked) them to turn on their stretcher beds to look the other way. . . zero privacy. . .this was wrong but I had no choice . . . I apologised to the mother. (mw15)*

*A woman was screaming . . . the ward was busy . . .the (other) midwives took no notice. . .I clamped and cut the cord and put the baby (I had just delivered) comfortably on the mother's chest then ran to the next cubicle. . . the mere sight of the baby on the floor . . .made me feel powerless, helpless, anxious, nervous, guilty and ashamed. . .We discussed this incident*

*afterwards and encouraged each other to (listen) to mothers more and not regarding their cries or screams as attention seeking.(mw15)*

The midwives overcame their sense of shame by sharing these stories with each other through their community of practice, through finding they were not alone and isolated, but most importantly, by finding solutions together. Space for birthing women to have companions with them, or to walk around or adopt alternative positions, were constraints if the cubicles were crowded. Despite these limitations they did their best to collectively implement what they learnt.

*What a shift, it was busy. . . we would ask each other on the way forward if we met any complications. . . . . .we treated each patient as an individual with individual needs. (mw14)*

*We organised ourselves that we will not bump into each other. . .or leave women unattended. . . it was extremely busy . . .teamwork helped us to pull through. . .At the end of the night shift, was exhausted, tired and felt drained physically but not emotionally. . . happy we managed to serve all mothers without any complications. . . I could have felt frustration, anxious . . . I now understand how to control emotions at work and to go outside and do deep breathing if I am overwhelmed.* (mw16)

They became highly motivated to change what they could, especially in trying non-pharmaceutical methods of pain-relief, challenging conventions that prohibited the presence of birth companions, and forced women to give birth in supine positions, advocating for the rights of birthing women based on the RMC Charter [20].

**Evidence of increasing agency in midwives' advocacy initiatives.** In the follow-up meetings, the midwives reported how they were improving theirs' and their colleagues' professional practices and team culture by discussing their experiences and management of complex cases during team meetings. They arranged feedback meetings with their workplace supervisors and colleagues, to relay to them the lessons learnt during the training and to gain their support for the changes. Some included ancillary staff and cleaners who interfaced with women in labour. They reported that their teams committed to immediate no-cost actions such as improving communications and teambuilding, giving birthing women and their families updates during labour, more flexibility over birth companions, and encouraging movement and different positions during labour and childbirth. Posters on birthing positions and the RMC Charter on women's rights during pregnancy and childbirth were displayed in antenatal clinics and birthing rooms and used for education talks. The midwives got commitment from senior management to find resources to create spaces for labouring women to walk with companions (such as in corridors or hospital grounds), for midwives to attend meetings and for further RMC training. Monthly maternal and perinatal mortality meetings have been held for more than 30 years, attended by the whole range of health professionals involved in maternity services for the Greater Harare Maternity Unit (primary, secondary and tertiary care) [42]. The midwives from this study reported feeling confident to promote RMC at these meetings as a means of improving birth outcomes (with accompanying evidence), highlighting patient safety and the need for more focused allocation of resources.

The inclusion of RMC into pre-service midwifery curricula and as part of induction of newly qualified midwives, was endorsed. Midwives in public sector facilities resolved to get funding for further RMC educational meetings and for leaflets about positions, companions, breastfeeding and family planning, to give mothers and their companions during antenatal and postnatal care. Midwives in private facilities had more flexibility to make changes and

purchased birthing balls, birthing stools and floor mats, invited companions to antenatal classes for pregnant women and developed educational materials on roles and responsibilities for companions. Visiting times were extended and included children. They set up open days to show pregnant women and their companions around maternity units, where the various routines and procedures were explained. Through these activities the midwives showed enhanced and sustained agency, to undertake purposeful measures to achieve their goals.

## Reflexivity and trustworthiness

The method of longitudinal data collection, with checking back and adding to the data (member checking or participant validation) during the follow-up meetings and the one-year participative workshop, contributed to the trustworthiness of the analysis. The categories, themes, interpretations and conclusions of the data analysis were verified with the midwives, to confirm that they accepted the representation of their realities and experiences, asking whether they had amendments or modifications. To check for possible bias, an independent colleague peer-reviewed a random selection of five journals, giving comments on the main points arising from the journals, which compared well with the overall analysis of the journal reflections. We were alert to the tension between our roles in this study, that in facilitating the reflections and discussions reported by the midwives, we may have influenced their views and assumptions. To preserve the methodological rigour of the research, SR kept detailed notes of our interactions and the progression of ideas over the course of the programme. There were indeed key influences that led to positive outcomes. Witnessing our commitment and enthusiasm as facilitators, and that we cared about them and birthing women, the midwives felt safe to express the caring side of their professionalism. There may have been social desirability bias affecting the responses of the midwives, in that they gave answers that would be viewed favourably by the facilitators. Their narratives showed that they knew how to provide RMC, reinforced by the positive feedback they received.

The midwives' journal entries and workshop discussions explored the constraints they faced, with ideas on how to overcome them. The journal narratives they selected to read to the group occasionally exposed their disrespectful behaviour, but they were keen to hear how others would react in similar situations. They took responsibility for having shouted, scolded and neglected women in labour, to understand why they behaved that way and what was needed to change. Two midwives expressed "*shock*" in their journals at some of the behaviours described by their colleagues. Some confirmed their past behaviour: one said, "*I was that kind of midwife who would always shout and scold patients. . .*"; another said, "*I was embarrassed when I heard my colleague shouting at her patient in labour, because I knew that was how I also behaved before the RMC training. . . I did not feel I could talk to her about it because she would think I was acting superior to her*". During the two meetings (one year apart) where the analysis was presented to the group as part of the member-checking process, the midwives agreed that the analysis covered the main issues, with consistency in the events and responses described. A year after the training, the midwives described how they had sustained many of the changes they made despite the constraints they worked under, especially in the public sector.

## Discussion

This study presents the voices of midwives, describing their lived experiences of caring for women during facility-based childbirth and how they changed through an educational process using transformative learning and critical reflection techniques. Although the study involved a small group of selected midwives, it yielded important insights into some of the underlying reasons for disrespect towards birthing women, with lessons for more effective midwifery

training in low and middle-income countries (LMICs) and better-resourced settings. The midwives expressed gratitude to be heard. Their progression through transformative learning can be discerned from their reflections and discussions, as detailed in Table 1, starting with their descriptions of "disorienting dilemmas", of incidents that occurred in their workplaces. These incidents exemplified the harrowing and stressful circumstances the midwives faced every day. Transition through the sequence of phases following those descriptions was not linear, although this is not considered essential for transformation to take place [27,43,44].

The midwives progressed from describing their verbal abuse and neglect of birthing women to self-identifying as agents of change and advocates for improved maternity care. The participatory and collaborative interventions in this study gave the midwives safe spaces in which to challenge their prior assumptions, such as that teenage mothers were irresponsible, or that women were lazy and would not progress in labour if they relaxed. They went on to make sense of their experiences based on their revisited values, beliefs and history, to explore and build new roles and responsibilities, and to reintegrate new ways of working into their practices. They collectively reimagined how they saw themselves as health professionals, what that meant in terms of attitudes and behaviour, the relationships they wanted with each other and the women they served. They became advocates for practices that were more women-centred, looked for solutions to the constraints they faced of space and time to enable women to move around in labour and to have companions, believing the scientific evidence that these led to better childbirth outcomes. They developed more compassion towards women in labour, and in the process became more compassionate towards themselves and each other. As part of self-care, they practiced breathing exercises, singing and stretching, to keep calm during times of stress, purposefully supported each other in reducing emotional exhaustion, and encouraged each other to smile more and be kind.

The perceived lower status of midwifery relative to that of obstetricians and medical doctors within the deeply hierarchical structures of health systems in many LMICs, contributes to lower self-efficacy, with the expertise and skills of midwives in childbirth unrecognized. In this study some midwives complained that they were treated as "handmaidens", expected to only carry out instructions and not think for themselves, even though they are on the frontline of most maternity services. The RMC training encouraged the midwives to demonstrate higher autonomy in applying knowledge and skills to decision-making, alongside a stronger sense of competence and authority, and to benefit from greater relatedness to each other and birthing women. Becoming self-aware and patient-oriented, as manifestations of building competence and confidence in new roles and relationships, are key attributes of transformative learning. Midwives found the confidence and authority to encourage senior colleagues and doctors to behave more professionally, to respect their opinions and to be better team-players, negotiating better care for birthing women as their advocates, influencing the team culture to be more woman-centred. Aspects of professionalism in relation to midwifery have been less explored previously even though it is the foundation for RMC [15]. The collective construction of the midwives' positive professional identity was a key outcome of this study where self-determination played a vital role in facilitating professional pride and resilience in the midwives [23,24]. Identity development is described as "interpreting oneself as a certain kind of person, presenting oneself as that person and being recognised as such in a given context" [45:e212]. Trying out new roles and having ones' achievements recognised by others aligns with transformative learning and identity development.

Journal-writing and reflections were done individually, but exploration of their feelings, perceptions and behaviour change occurred through group-work during the follow-up sessions, as they established and reinforced in each other new ways of viewing the world and understanding each other. They continued as a community of practice after the RMC training

concluded, supporting each other in being more open, inclusive, reflective and emotionally able to change. The midwives spoke with pride of their roles and responsibilities as professional midwives, of the negotiation and conflict resolution skills they developed through the RMC training and through their own teambuilding during the year, as well as sustaining many of the changes they made in their practices. Their identities now included themselves as champions and change agents, actively promoting attainment of human rights for women in terms of RMC.

Using a human rights approach to maternal health, including promotion of the RMC Charter [20], created a light bulb moment of realization and inspiration for the midwives, that women had the right to dignity, privacy, information, freedom from harm, to have companions, to choose their positions for giving birth. This revelation acted to put midwives on a more equal footing with women in labour, to reduce the social distance between them and to engage with them as adults. The midwives learnt through their reflections and dialogues what lectures on professionalism could not have achieved. They revisited their assumptions with new perspectives, reversing their approaches towards women with difficulties, understanding more what they were going through by listening to them, instead of distancing from them or abusing them [15,16]. They learnt that resistance from birthing women was often because of fear of the unknown, of pain and danger. These changes were dramatically witnessed with younger unmarried women, for whom negative attitudes of health workers often act as barriers to good sexual health [46]. as though their misfortunes are deserved as punishment for breaking the rules. The change towards greater compassion in the midwives' responses, giving advice and encouraging further care and support, meant that these young women were more likely to have better reproductive health in future. In treating birthing women with respect, providing information so that they also had choices and autonomy, listening more to what their needs were and acting on them, the midwives demonstrated greater respect and compassion for themselves and each other. The appreciative feedback they received from birthing women and colleagues reinforced their sense of professionalism, competence and relatedness, as did the support they received from senior health professionals and managers. Focusing on what went well rather than only what went wrong, helped to boost morale and provided ideas of what could be done better.

Although failures in maternal health services underlay many of the stresses faced by the midwives, they took ownership of the small incremental changes they could make to improve women's birthing experiences, despite the long hours they worked in difficult conditions. They focused on their immediate relationship with women in labour rather than being defeated by the social and economic difficulties faced by the women they served. The midwives' narratives revealed how they overcame their loss of morale, helplessness and shame by seeing themselves as change agents and RMC champions. This observation provides an alternative perspective to the suggestion that training midwives to manage their constraints more effectively burdens them with the responsibility to cope, rather than challenging the health system for its deficiencies [15]. Many public sector health facilities in less-resourced environments struggle to improve quality of care in the face of workforce shortages and burn-out, inadequate facilities and lack of equipment. Advocacy to address such health system weaknesses must continue alongside ways to use existing resources more effectively. Many of the interventions proposed within RMC can be carried out with minimal financial inputs, or with adaptations of existing equipment and space, but have been shown to have considerable impact on staff morale and enthusiasm, and patient satisfaction. Encouraging birth companions and use of alternative birthing positions may need more space especially in busy labour wards, but the midwives, in their mindsets as change agents, looked for solutions "outside the box" and campaigned for these with their managers.

A concern of moving from research to action is whether the changes achieved would be sustained outside of the study environment without the resources and influences that usually accompany research. The challenge is whether the midwives changed sufficiently to practice RMC without the workshops and continuing facilitation. Learning experiences are said to be transformative when the changes claimed persist and endure, when they are not limited to behaviours but to a way of seeing the world, being more open-minded, inclusive and having deeper self-awareness [43,47,48]. They are then likely to be relevant and connected across learning (integrative) and unlikely to be forgotten (irreversible) [16]. Whether the changes described by the midwives in this study are truly irreversible and integrative will only be seen through future reviews of their work. Similar methodologies of guided reflective writing and facilitated follow-up discussions would be required for new learners with the RMC champion midwives trained as facilitators and mentors. Health professionals need safe spaces to express themselves, to interact and to develop agency in addressing hierarchical issues present between them, and between them and the people they serve [34]. The RMC workshops created dedicated spaces, away from each person's workplace, where the midwives felt free to take part in critical reflection, even exposing their earlier weaknesses. Whether such training can be replicated within workplaces where this freedom may not be available needs further research.

To measure impact of the midwives' behaviour and practices in improving health outcomes will need alternative study designs, such as patient satisfaction surveys, monitoring indicators of quality of care and quality improvement studies. At present, it seems that peer support and approval from managers is sustaining these midwives in their new roles as change agents, especially in their team meetings. The focus of this paper has been on the narratives of the midwives, foregrounding their perceptions and experiences, emphasising the value of innovative training methods. We recognise however that the structural and policy related issues that influence the spaces within which the midwives work and the power dynamics that are at play in these spaces, have not gone away. Further research is needed to show how RMC can be embedded and sustained in organizational and team cultures, in flattening professional hierarchies operating in maternity services and promoting interprofessional collaboration, how health policies and payment systems can be used to improve the working environment for maternity teams [17,18]. Research is also needed on how civil society organisations, advocacy groups and professional associations can hold health services, governments, and policy-makers to account for inadequate maternity care and poor birth outcomes also [49].

The features of this study using transformative learning theory to support midwives in becoming meaningful RMC change agents were: professionalism and agency; being women-centred and responsive to needs; use of scientific knowledge to inform practice; critical reflective reasoning skills; non-hierarchical relationships and effective team-work; collaborative decision-making; communication skills; local resource mobilization and sharing. The challenge is whether it is only through this intensive process of transformative learning that RMC approaches can be instilled in midwifery training. There are models of training that have incorporated RMC and these educational methods into midwifery curricula. Educational tools have been proposed to facilitate reflective skills in midwifery students [50], to integrate theoretical knowledge with practice and learning from powerful experiences and interactions underlying observable activities in work [35,51]. Skilled facilitation by educators is essential to support learners in situations that may be distressing or uncertain, offering dilemmas to learners to provoke curiosity and exchange of perspectives, highlighting the complexities and ambiguities learners have to respond to [30,31,34]. At least one midwifery training programme in Zimbabwe has included RMC into its curriculum using these methods (P Mataure, personal communication). Short courses and workshops encouraging flexibility in birthing positions and involvement of birth companions have been held in other hospitals. A toolkit based on

our workshop materials has been produced by this team for use as training aides, which has been shared widely. Future research will reveal how effective these methods have been.

## Conclusion

The educational intervention at the centre of this study was different from conventional midwifery training. It was centered in self-reflection and dialogue, reframing key assumptions and progressing to self-awareness and confidence, positive professional identities, with greater compassion for women in labour and enhanced relatedness with colleagues. Lectures and didactic information dissemination are unlikely to have the same impact. The midwives acquired skills that enabled them to continue their reflective exercises, incorporating lessons from these into team-building, weekly feedback meetings and advocacy for resources. Collective imagining, dialogue and action are vital components of this method, with authentic engagement, support during times of stress and disillusion, shared experiences and knowledge, leading to motivation for change (30). This learning method is resource-intensive and requires considerable commitment from learners and facilitators. This study explores what is needed for sustained changes in attitudes, behaviour and practices in a group of midwives. It also demonstrates the power of narratives in understanding their interaction with the health system. With the ultimate goal of improving quality of care, health outcomes and birth experiences for women, the study suggests ways of achieving meaningful culture changes in midwifery training in sustainable and enduring ways, using innovative methods of promoting effective group working, skills for relationship building, and support networks to carry them through the tough times to the rewards and benefits that follow.

## Acknowledgments

This paper is based on a research report submitted by SR as part fulfilment for her MPhil HPE, Centre for Health Professions Education, Stellenbosch University, South Africa.

The authors give thanks to the White Ribbon Alliance for Safe Motherhood Zimbabwe andWomen's University Zimbabwe for their partnership with the workshops and production of the toolkits; to the midwives who participated in the educational intervention and study, to the Ministry of Health & Child Care, Sally Mugabe Central Hospital and the private maternity services for permission to conduct this study; and to Vicky Owens and Sarah Fox, midwives from Swansea Bay University Health Board Wales, for facilitation during the workshops.

## Author Contributions

**Conceptualization:** Sunanda Ray, Christina Mudokwenyu-Rawdon, Myriam Bonduelle, Ginny Iliff, Caroline Maposhere, Cecilia Jacobs, Susan C. Van Schalkwyk.

**Data curation:** Sunanda Ray, Cecilia Jacobs, Susan C. Van Schalkwyk.

**Formal analysis:** Sunanda Ray, Cecilia Jacobs, Susan C. Van Schalkwyk.

**Funding acquisition:** Sunanda Ray, Myriam Bonduelle.

**Investigation:** Sunanda Ray.

**Methodology:** Sunanda Ray, Christina Mudokwenyu-Rawdon, Ginny Iliff, Caroline Maposhere, Priscilla Mataure, Cecilia Jacobs, Susan C. Van Schalkwyk.

**Project administration:** Sunanda Ray, Christina Mudokwenyu-Rawdon, Myriam Bonduelle, Ginny Iliff, Caroline Maposhere, Priscilla Mataure, Susan C. Van Schalkwyk.

**Resources:** Sunanda Ray, Myriam Bonduelle, Ginny Iliff, Priscilla Mataure, Cecilia Jacobs, Susan C. Van Schalkwyk.

**Supervision:** Sunanda Ray, Caroline Maposhere, Priscilla Mataure, Cecilia Jacobs, Susan C. Van Schalkwyk.

**Validation:** Sunanda Ray, Caroline Maposhere, Cecilia Jacobs, Susan C. Van Schalkwyk.

**Visualization:** Sunanda Ray.

**Writing – original draft:** Sunanda Ray, Myriam Bonduelle, Ginny Iliff, Caroline Maposhere, Priscilla Mataure, Susan C. Van Schalkwyk.

**Writing – review & editing:** Sunanda Ray, Christina Mudokwenyu-Rawdon, Myriam Bonduelle, Ginny Iliff, Caroline Maposhere, Priscilla Mataure, Cecilia Jacobs, Susan C. Van Schalkwyk.

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
