## [Decision Letter · Decision Letter 0]

5 Jun 2023

PGPH-D-23-00084

Hearing the voices of midwives through reflective writing journals: qualitative research on an educational intervention for Respectful Maternity Care in Zimbabwe

Dear Dr. Ray,

Thank you for submitting your manuscript to PLOS Global Public Health. After careful consideration, we feel that it has merit but does not fully meet PLOS Global Public Health’s publication criteria as it currently stands. Therefore, we invite you to submit a revised version of the manuscript that addresses the points raised during the review process.

Please note that we have only been able to secure a single reviewer to assess your manuscript. We are issuing a decision on your manuscript at this point to prevent further delays in the evaluation of your manuscript. Please be aware that the editor who handles your revised manuscript might find it necessary to invite additional reviewers to assess this work once the revised manuscript is submitted. However, we will aim to proceed on the basis of this single review if possible.

We look forward to receiving your revised manuscript.

Kind regards,

Jianhong Zhou

Staff Editor

Journal Requirements:

Additional Editor Comments (if provided):

Reviewers' comments:

Reviewer's Responses to Questions

**Comments to the Author**

1. Does this manuscript meet PLOS Global Public Health’s publication criteria? Is the manuscript technically sound, and do the data support the conclusions? The manuscript must describe methodologically and ethically rigorous research with conclusions that are appropriately drawn based on the data presented.

Reviewer #1: Yes

2. Has the statistical analysis been performed appropriately and rigorously?

Reviewer #1: N/A

3. Have the authors made all data underlying the findings in their manuscript fully available (please refer to the Data Availability Statement at the start of the manuscript PDF file)?

Reviewer #1: No

4. Is the manuscript presented in an intelligible fashion and written in standard English?

Reviewer #1: Yes

5. Review Comments to the Author

Reviewer #1: Thank you so much for taking on this research on a very important issue and also for selecting unique methods to explore this topic. I have some generic and specific comments that could help make this manuscript stronger.

Generic comments-

• Language- Please don’t refer to women and other people in labour and birth as ‘patients’. Please also change ‘delivery or delivering’ to ‘giving birth’ unless it is in a quote.

• Methods- would be good to get more details on the journals, the language, the challenges in analysis etc.

• Positionality- Were there any midwives or nurse-midwives amongst the co-authors? Please include a little more about the co-authors.

• ‘Compassion’ and ‘empathy’ have been used interchangeably in the manuscript. Please look into the meaning of and differences in these two and use the concepts accordingly.

Specific comments

Page 15- last paragraph- Behaviour change is not as simple as suggestions from peers to smile more and be kind. Please look back into the journals to find more on what makes them loose that smile, loose their compassion and be unkind in the first place. What are the structural and policy related barriers and gender and power related issues that makes RMC challenging in obstetric settings? There is potential to add a sub-section about this in the findings to then understand this better.

Page 16- 1st paragraph about midwives’ newfound confidence and authority- While this has been mentioned in the findings as well. It is not as easy. What made the challenges / barriers that they may have faced before go away? what are the repercussions of them exercising this newfound agency? if they are not understood better then chances are that with time the old team culture will catch up.

Last paragraph about uncooperative children- I understand that this language of 'disciplining naughty children' has been referenced to in the introduction as well. But literature suggests that this is also gender -based and there is plenty of evidence of gender-based violence being about disciplining women and obstetric violence is one of it’s kind as well. The literature around women self-disciplining their behaviour and bodies to avoid obstetric violence also points to that. Please bring that into the discussion.

Page 17- 2nd paragraph about midwives’ helplessness- Exactly! More on this in the findings from the midwive's journals please.

Maternal morality meetings- It is unclear to me whether these meetings are being attended by just midwives or the entire team of care providers.

Page 18- 1st paragraph about future research- Please bring in suggestions that are about changing the team culture and not just focused on the midwives.

Conclusion-

Page 19- From what I understand, this study explores this only in the context of midwives not all health care professionals. Please change the sentence. Every category of health care professional's challenges are unique, so it won't be fair to generalize this to health health care providers.

References- please provide the doi of all the references where applicable.

6. PLOS authors have the option to publish the peer review history of their article (what does this mean?). If published, this will include your full peer review and any attached files.

**Do you want your identity to be public for this peer review?** For information about this choice, including consent withdrawal, please see our Privacy Policy.

Reviewer #1: **Yes: **Kaveri Mayra

---

## [Editor Report · Decision Letter 1]

28 Nov 2023

Hearing the voices of midwives through reflective writing journals: qualitative research on an educational intervention for Respectful Maternity Care in Zimbabwe

PGPH-D-23-00084R1

Dear Dr Ray

We are pleased to inform you that your manuscript 'Hearing the voices of midwives through reflective writing journals: qualitative research on an educational intervention for Respectful Maternity Care in Zimbabwe' has been provisionally accepted for publication in PLOS Global Public Health.

Best regards,

Gizachew Tessema, PhD

Academic Editor